# Cytokine-Induced Myeloid-Derived Suppressor Cells Demonstrate Their Immunoregulatory Functions to Prolong the Survival of Diabetic Mice

**DOI:** 10.3390/cells12111507

**Published:** 2023-05-29

**Authors:** Tung-Teng Li, Chun-Liang Lin, Meihua Chiang, Jie-Teng He, Chien-Hui Hung, Ching-Chuan Hsieh

**Affiliations:** 1Division of General Surgery, Chang-Gung Memorial Hospital, Chiayi 61302, Taiwan; b9702020@cgmh.org.tw (T.-T.L.); maha0210@gmail.com (M.C.); r90225029@ntu.edu.tw (J.-T.H.); 2Department of Nephrology, Chang-Gung Memorial Hospital, Chiayi 61302, Taiwan; linchunliang@cgmh.org.tw; 3Kidney and Diabetic Complications Research Team (KDCRT), Chang-Gung Memorial Hospital, Chiayi 61302, Taiwan; 4College of Medicine, Chang-Gung University, Taoyuan 33302, Taiwan; hungc01@mail.cgu.edu.tw; 5Division of Infectious Diseases, Chang-Gung Memorial Hospital, Chiayi 61302, Taiwan

**Keywords:** cytokine, myeloid-derived suppressor cells, diabetic nephropathy, pancreatic insulitis, immunotherapy

## Abstract

Type 1 diabetes is an inflammatory state. Myeloid-derived suppressive cells (MDSCs) originate from immature myeloid cells and quickly expand to control host immunity during infection, inflammation, trauma, and cancer. This study presents an ex vivo procedure to develop MDSCs from bone marrow cells propagated from granulocyte–macrophage-colony-stimulating factor (GM-CSF), interleukin (IL)-6, and IL-1β cytokines expressing immature morphology and high immunosuppression of T-cell proliferation. The adoptive transfer of cytokine-induced MDSCs (cMDSCs) improved the hyperglycemic state and prolonged the diabetes-free survival of nonobese diabetic (NOD) mice with severe combined immune deficiency (SCID) induced by reactive splenic T cells harvested from NOD mice. In addition, the application of cMDSCs reduced fibronectin production in the renal glomeruli and improved renal function and proteinuria in diabetic mice. Moreover, cMDSCs use mitigated pancreatic insulitis to restore insulin production and reduce the levels of HbA1c. In conclusion, administering cMDSCs propagated from GM-CSF, IL-6, and IL-1β cytokines provides an alternative immunotherapy protocol for treating diabetic pancreatic insulitis and renal nephropathy.

## 1. Introduction

Type 1 diabetes (T1D) is characterized by a chronic inflammatory response associated with the autoimmune destruction of insulin-producing beta cells in the pancreas [1]. Emerging evidence has revealed that autoreactive T cells play a crucial role in spontaneous pancreatic insulitis resulting in β-cell loss [2]. Complications of T1D include renal failure, retinopathy, ischemic heart disease, and chronic foot ulcers [3]. 

Myeloid-derived suppressive cells (MDSCs) originate from immature myeloid cells and quickly expand to regulate the host’s immune response during cancer, trauma, infection, and some autoimmune disorders [4,5]. In mice, MDSCs are characterized by the coexpression of the myeloid lineage differentiation antigen Gr1 and CD11b [6]. In humans, MDSCs are broadly defined as CD14^−^CD11b^+^ cells or cells that express CD33 but lack mature myeloid and lymphoid cells markers and the MHC class II molecule HLA-DR [7,8]. The suppressive activity of MDSCs has been associated with arginase I, inducible nitric oxide synthase (iNOS) [9,10], reactive oxygen species [11,12], anti-inflammatory cytokines [5], regulatory T cells [13,14], and tumor-associated macrophages [15,16].

Immunotherapy is a method of disease treatment that involves stimulating or suppressing the immune response and provides an alternative treatment protocol for patients with cancer, inflammation, and organ transplantation [17]. The anti-inflammatory and immunoregulatory properties of MDSCs are commonly used in immunotherapy [18]. The standard protocol is to use MDSCs differentiated from peripheral blood mononuclear cells or bone marrow (BM) cells with high concentrations of granulocyte–macrophage-colony-stimulating factor (GM-CSF) by itself or in combination with other factors [19]. 

This study propagated MDSCs from mouse BM cells with GM-CSF, interleukin (IL)-6, and IL-1β cytokines in vitro and characterized their functions. In addition, we examined the regulatory effects of cytokine-induced MDSCs (cMDSCs) on pancreatic insulitis and renal glomerulosclerosis in nonobese diabetic (NOD) mice and mice with severe combined immune deficiency (SCID).

## 2. Materials and Methods

### 2.1. Mouse Models

Female BALB/c, C57BL/6 (B6), NOD, and NOD–SCID mice were supplied by the National Laboratory Animal Center (Taiwan). All animal experiments were approved by the Institutional Animal Care and Use Committee of Chang Gung Memorial Hospital (IACUC permit number: 2016032802) and were performed following the Animal Protection Law by the Council of Agriculture, Executive Yuan (Taiwan), and the National Research Council’s Guide for the Care and Use of Laboratory Animals (USA). 

### 2.2. Culture of cMDSCs 

BM cells (2 × 10^6^ cells/well) from the tibias and femurs of the BALB/c mice were cultured in Roswell Park Memorial Institute 1640 medium containing 10% fetal bovine serum with mouse recombinant GM-CSF (10 ng/mL; R&D Systems, Minneapolis, MN, USA), IL-6 (10 ng/mL; R&D Systems, Minneapolis, MN, USA), and IL-1 β (10 ng/mL; R&D Systems, Minneapolis, MN, USA) for 7 days. Cells coexpressing CD11b and Gr-1 were considered cMDSCs. 

### 2.3. cMDSCs Cotransplanted with Immunized Splenic T Cells into NOD–SCID Mice

T cells (CD3^+^, 2 × 10^7^ cells) harvested from the spleens of NOD mice were injected intravenously to the NOD–SCID mice with or without the cMDSCs (1 × 10^7^ cells) twice a week for 5 consecutive weeks. The total adoptive cell volume for intravenous injection was 20 million cells in the group of T cells alone (*n* = 5) and 30 million cells in the group of T cells cotransplanted cMDSCs within 100 ul every time (*n* = 5). Blood sugar levels and body weight were measured weekly. Diabetes was diagnosed with a sustained blood sugar level > 350 mg/dL using a OneTouch UltraEasy monitor (Johnson and Johnson, New Brunswick, NJ, USA).

### 2.4. Flow Cytometry Analysis

Monoclonal antibodies (mAbs) against CD11b, CD40, CD86, Gr-1, and MHC class II (I-A^d^), were purchased from BD PharMingen (San Diego, CA, USA), and mAbs against B7-H1 were purchased from eBioscience (San Diego, CA, USA). For CFSE labeling, splenic T cells (10^7^ cells/mL) from B6 were incubated with 0.5 μM of CFSE (Invitrogen, San Diego, CA, USA) for 10 min at room temperature. Flow analyses were performed using a BD FACSCanto II flow cytometer (BD Bioscience, Franklin Lakes, NJ, USA).

### 2.5. Quantitative Reverse Transcription Polymerase Chain Reaction

Total RNA was extracted using an RNeasy Mini Kit (Qiagen, Valencia, CA, USA). RNA samples were converted into cDNA using the RevertAid First Strand cDNA Synthesis Kit (Thermo Fisher, Waltham, MA, USA). The following primers were used for the quantitative polymerase chain reaction (qPCR): arginase 1: forward CACGG CAGTG GCTTT AACCT and reverse TGGCG CATTC

ACAGT CACTT; iNOS: forward TGGCC ACCTT GTTC AG CTACG and reverse GCCAA GGCCA AACAC AGCAT AC; fibronectin: forward GCTCA GCAAA TCGTG CAGC and reverse CTAGG TAGGT CCGTT CCCACT; STAT1: forward CTGAATATTTCCCTCCTGGG and reverse TCCCGTACAGATGTCCATGAT; STAT3: forward CTTGTCTACCTCTACCCCGACAT and reverse GATCCATGTCAAACGTGAGCG. mRNAs were measured using the CFX96 Touch Real-Time PCR Detection System (Bio-Rad Laboratories, Inc., Hercules, CA, USA) in duplicate and normalized to 18S mRNA.

### 2.6. Immunofluorescence Staining and Confocal Microscopy

Tissue samples of pancreatic islets were embedded in optimal cutting temperature compound and snap-frozen in liquid nitrogen. Tissue sections (4 μm) were fixed in ethanol–acetic acid fixative solution for 2–10 min and stained with anti-insulin antibody (1:100, SC-8033, Santa Cruz Biotechnology, Inc., Santa Cruz, CA, USA) with secondary antibody (1:400, Fab’2 Donkey anti-Mouse IgG-AF594, Jackson ImmunoResearch Inc. West Grove, PA, USA), DAPI (Invitrogen, Waltham, MA, USA), and Alexa Fluor 488 Phalloidin (Invitrogen, Waltham, MA, USA) overnight at 4 °C in a humidified chamber. After three washes, slides were stained with DAPI and mounted with ProLong Gold mounting medium (Invitrogen, Waltham, MA, USA). Confocal imaging was performed using a Leica SP5 II confocal microscope.

### 2.7. Immunohistochemistry

Extracellular matrix expression of the renal cortex in the cryostat sections was identified through fluorescent staining using specific antifibronectin (Abcam, Cambridge, UK), anti-insulin (Santa Cruz Biotechnology, Inc., Santa Cruz, CA, USA), anti-CD3 (BD PharMingen, San Diego, CA, USA), and anti-Gr1 (BD Biosciences, Franklin Lakes, NJ, USA) antibodies after permeabilization with 0.05% saponin buffer using a Vectastain Elite ABC kit (Vector Lab, Inc., Burlingame, CA, USA) for an immunoperoxidase procedure. The slides were developed using AEC chromogen substrate and counterstained with hematoxylin. Isotype- and species-matched irrelevant antibodies served as controls.

### 2.8. Blood and Urine Analysis

Blood glucose was determined using a OneTouch UltraEasy monitor (Johnson and Johnson, New Brunswick, NJ, USA). Blood and urine were collected to examine serum creatinine, HbA1c, and urinary protein when the mice were sacrificed. Serum creatinine, HbA1c, and urinary protein levels were determined using a Labospect 008 (Hitachi, Tokyo, Japan).

### 2.9. Statistical Analysis

The statistical analysis was performed using Student’s *t*-test for independent samples, with *p* < 0.05 indicating significance. A Mann–Whitney U test was performed to analyze diabetes-free survival. All data median and range were calculated and graphed in Microsoft Excel 2016 version 1.0 (Microsoft, Redmond, WA, USA).

## 3. Results

### 3.1. Spontaneous Pancreatic Insulitis and Renal Glomerulosclerosis in NOD Mice

The blood sugar levels of the NOD mice at 7 weeks old were 126.0 ± 4.0 mg/dL and gradually increased to higher than 350 mg/dL at the age of 10 weeks (358.0 ± 2.5 mg/dL). The blood sugar levels of the BALB/c mice remained steady at approximately 116.0 ± 2.6 mg/dL throughout the experiment (Figure 1a, left panel). Decreases in the body weight of the NOD mice were associated with hyperglycemia, whereas slight increases in the body weight of the BALB/c mice were not associated with blood sugar levels (Figure 1a, right panel). 

Autoimmune diabetes spontaneously developed in the NOD mice from autoreactive T cells. Hematoxylin and eosin staining of the pancreatic islets of the NOD 14-week-old mice revealed that, compared with the pancreatic islets of the BALB/c mice, the islet cell mass of the NOD mice was lost, and the islets were surrounded by leukocytic infiltration (Figure 1b, upper-left panel). A substantial decrease in insulin production due to beta cell mass loss was observed in the NOD mice through immunofluorescence staining (Figure 1b, upper-right and lower-right panels). Blood glycosylated hemoglobin (HbA1c) levels (%) in the NOD mice were higher than those in the BALB/c mice (Figure 1b, lower left panel). 

Nephropathy is a major complication of diabetes. Extracellular matrix (ECM) accumulation in the renal parenchyma is a key etiological element in the pathogenesis of diabetic nephropathy. In our study, fibronectin, a major component of ECM, was more highly expressed in the renal glomerulus of the NOD mice than in that of the BALB/c mice (Figure 1c, upper panels). Serum creatinine (0.81 ± 0.10 vs. 0.33 ± 0.10 mg/dL) and urinary protein (196.67 ± 16.0 vs. 31.0 ± 3.6 mg/dL) levels were noticeably higher in the NOD mice than in the BALB/c mice (Figure 1c, lower panels). 

Diabetes occurred in the NOD mice due to spontaneous insulitis, leukocytic infiltration within pancreatic islets, leading to the loss of pancreatic beta cells. Renal dysfunction occurred after fibronectin accumulated in the renal glomerulus, resulting in glomerulosclerosis during hyperglycemia. 

### 3.2. MDSCs Differentiated from GM-CSF, IL-6, and IL-1β Cytokines Displayed Potent Immunosuppressive Activity

MDSCs are propagated from myeloid progenitor cells and exhibit immunosuppressive activity rather than immunostimulatory properties. To ensure the strength of the immunosuppressive properties of the MDSCs, we developed several ex vivo procedures to differentiate MDSCs from BM cells through the GM-CSF cytokine by itself or in combination with other molecules. The results displayed in Figure 2a reveal that the yield ratio of MDSCs was higher in the G + IL-6 + IL-1β group (92.6%) than in the GM-CSF group (54.6%), the G + IL-6 group (78%), and the G + IL-1β group (73%; Figure 2a, upper-left panel). Among the four groups, the production of MDSCs was also highest in the G + IL-6 + IL-1β group (Figure 2a, right panel). The MDSCs propagated from the G + IL-6 + IL-1β group were round and exhibited a small amount of cytoplasmic projection and a large nucleus-to-cytoplasm ratio, revealing the immature state of the immune cells (Figure 2a, lower-left panel). The MDSCs in the G + IL-6 + IL-1β group expressed fewer costimulatory CD40, CD86, and MHC class II molecules and more inhibitory B7H1 among the four groups, indicating that MDSCs display a greater capacity to suppress the immune response under the influence of three mixed cytokines (Figure 2b).

In general, MDSCs obtain their immunosuppressive properties by expressing high arginase 1 and inducible NOS levels through the signal transducer and activator of transcription (STAT) family of transcription factors. In this study, MDSCs were propagated from GM-CSF, IL-6, and IL-1β cytokines that expressed considerably high levels of STAT1 but not STAT3 transcription factors in all four groups (Figure 2c, upper panel). Moreover, the MDSCs noticeably upregulated the expression of the iNOS enzymes rather than the production of arginase 1 under the influence of the GM-CSF, IL-6, and IL-1β cytokines (Figure 2c, lower panel). These results demonstrate that the MDSCs developed from the GM-CSF, IL-6, and IL-1β cytokines exuded their immunoregulatory properties through the iNOS enzyme via the STAT1 signaling pathway. 

To examine the effect of the MDSCs on adaptive immunity, the MDSCs were cocultured with allogeneic spleen T cells stained with carboxyfluorescein succinimidyl ester (CFSE), and the proliferation rate of the T cells was examined. The MDSCs from the G + IL-6 + IL-1β group exhibited the most potent ability to suppress the T-cell proliferation rate with a dose-dependent effect compared to the other groups (Figure 2d). IFNγ, a proinflammatory cytokine, was secreted less from T cells stimulated by GM-CSF, IL-6, and IL-1β molecules (Figure 2e). 

In conclusion, the MDSCs differentiated from the GM-CSF, IL-6, and IL-1β cytokines had immature morphology, upregulated the production of iNOS through STAT1 signaling, and exhibited high immunosuppression of T-cell proliferation. 

### 3.3. Application of cMDSCs Reduced Renal Fibronectin Production and Improved Pancreatic Insulitis to Prolong Diabetes-Free Survival

Autoreactive T cells prompt T1D development by causing severe destruction of pancreatic β cells. Studies have noted that MDSCs employ their immunoregulatory properties to suppress T-cell proliferation. To determine whether cMDSCs propagated from GM-CSF, IL-6, and IL-1β cytokines influence the properties of T cells that cause T1D diabetes in vivo, T cells were harvested from the spleens of NOD mice and intravenously administered to the NOD–SCID mice by themselves or cotransplanted with cMDSCs twice a week for 5 consecutive weeks (Figure 3a, upper panel). Blood sugar levels in the group treated with T cells alone were the highest among the three groups, whereas the blood sugar levels decreased in those treated with T cells cotransplanted with cMDSCs. The body weight of the NOD–SCID mice was lowest in the group treated with T cells alone, whereas the group treated with the combination of T cells and cMDSCs exhibited a slight decrease in body weight compared with the untreated group (Figure 3a, lower panel). Treatment with the cMDSCs cotransplanted with T cells considerably improved the diabetes-free survival of the NOD–SCID mice compared to the group treated with T cells alone (Figure 3b). As shown in Figure 3c, the population of MDSCs in the T cells + cMDSCs group was higher in the pancreas than in the other two groups.

Nephropathy causes the accumulation of ECM proteins in the mesangial interstitial space and is detrimental to diabetes. Treatment with T cells considerably increased the fibronectin production in the renal glomerulus of the NOD–SCID mice, and administration of cMDSCs improved fibronectin expression in the renal glomerulus (Figure 3d, upper panel). The serum creatinine and urinary protein levels noticeably increased in the NOD–SCID mice after treatment with T cells compared with the untreated mice, whereas the application of cMDSCs improved renal function and proteinuria in the NOD SCID mice (Figure 3d, lower panel). 

A deficiency of insulin production in pancreatic islets is the primary etiology of T1D. The data in the upper panel of Figure 3e demonstrate that insulin production in the NOD–SCID mice treated with the T cells harvested from NOD mice was the lowest among the three groups, whereas insulin production increased after cotransplantation with cMDSCs. HbA1c levels, the gold standard for monitoring chronic high blood sugar, were considerably increased in the group treated with T cells alone compared with the untreated group (Figure 3e, lower panel). The treatment of cMDSCs cotransplanted with T cells reduced HbA1c levels compared with the treatment of T cells alone. The upper panel of Figure 3F indicates that the number of T cells (CD3^+^) within the pancreatic islets was higher in the NOD–SCID mice treated with T cells alone or with cotransplanted cMDSCs than that in the untreated mice. The decrease in the number of T cells was correlated with the increase in the number of cMDSCs in the pancreatic islets of the NOD–SCID mice (Figure 3f, lower panel). The levels of IFNγ secreted from T cells harvested from pancreatic tissue were highest in the group of T cells alone among the three groups. The production of IFNγ was considerably reduced in the group of T cells cotransplanted cMDSCs compared to the group of T cells alone (Figure 3g). 

In summary, the application of cMDSCs propagated from GM-CSF, IL-6, and IL-1β cytokines can improve renal function and increase insulin secretion from pancreatic islets, thereby prolonging the diabetes-free survival of NOD–SCID mice after treatment with T cells (Figure 3h).

## 4. Discussion

MDSCs originate from the myeloid cell lineage and contain a diverse population of precursors of myeloid cells and myeloid cell progenitors. These immature myeloid cells quickly expand to control host immunity during infection, inflammation, trauma, and cancer [4,5]. The anti-inflammatory and immunoregulatory properties of MDSCs include increasing the expression of reactive oxygen species [11,12], arginase 1, iNOS [9,10], anti-inflammatory cytokines [5], regulatory T cells [13,14], macrophages associated with tumors [15,16], proangiogenic factors [20], and the impairment of natural killer cell cytotoxicity [21]. In a prior study, we demonstrated that MDSCs induced by hepatic stellate cells (HSCs) exhibited high immunoregulatory properties and prolonged the survival of mice with cotransplanted islet allografts through the induction of effector T-cell apoptosis and the generation of regulatory T cells [22]. The soluble molecule complement component 3 (C3) secreted from the HSCs was involved in the MDSCs’ expression of an inhibitory immune response [23]. In addition, the MDSCs produced inhibitory enzymes through the IL-6 signaling pathway to reduce effector T-cell immunity and promote tumor progression within the tumor microenvironment [24,25,26,27]. The role of MDSCs in inflammatory and infectious disorders has been studied [28,29].

Immunotherapy is a treatment method through the induction, augmentation, or inhibition of an immune response. Adoptive cell immunotherapy is an inspiring area of cancer therapy that commenced in the 1980s. Chimeric antigen receptor (CAR) T-cell immunotherapy for treating B-cell hematologic neoplasms is one of the most promising examples. Alternatively, to promote the further application of CAR T cells, natural killer (NK) cells or macrophages could be genetically transduced to generate CAR NK cells or CAR M cells that could be promptly implemented to non-cancer fields such as infectious disorders and autoimmune diseases [30,31]. Although the spleens and blood of tumor-bearing mice are the most commonly used sources of MDSCs for immunotherapy, many safety precautions must be considered [18]. The most widely used ex vivo protocol to propagate MDSCs from BM cells or peripheral blood mononuclear cells involves high concentrations of GM-CSF alone or in combination with other factors [19]. Our previous research revealed that IL-6 cytokine could increase the propagation of MDSCs within tumor microenvironments [24,26,27]. We reported that IL-1β cytokine derived from pancreatic cancer promoted the population of M2 macrophages, MDSCs, CD1d^hi^CD5^+^ regulatory B cells, and Th17 cells, resulting in immunoregulation [32]. Lechner et al. reported that MDSCs propagated in vitro from human peripheral blood mononuclear cells from GM-CSF and IL-6 exhibited the strongest ability to inhibit T-cell proliferation among seven cytokine mixtures [33]. In the present study, we developed an ex vivo procedure to develop MDSCs from BM cells propagated from GM-CSF, IL-6, and IL-1β cytokines that expressed less costimulatory and more inhibitory surface markers as well as considerably high levels of the iNOS enzyme through the STAT1 signaling pathway. In addition, the MDSCs differentiated from the GM-CSF, IL-6, and IL-1β cytokines exhibited immature morphology and strong immunosuppression of T-cell proliferation. 

T1D is an insulin-dependent disorder characterized by a chronic inflammatory state against insulin autoantigens, leading to the eventual destruction of pancreatic β cells [1]. Autoreactive T cells cause the loss of β cells associated with pancreatic insulitis [2]. NOD mouse models have been widely used and have been a crucial part of research on the mechanism of autoimmunity and T1D [34,35,36]. In this study, diabetes onset, that is, a blood sugar level > 350 mg/dL, began at 10 weeks of age in the NOD mice, accompanied by a decrease in body weight. Four weeks later, pancreatic islets were surrounded by leukocytic infiltration, leading to loss of islet cell mass. The effects of spontaneous pancreatic insulitis resulted in decreased insulin production and elevated levels of HbA1c in the blood. Nephropathy is a complication of diabetes that is the most common cause of renal failure worldwide [37]. Fibronectin, a major component of ECM, was highly expressed and accumulated in the renal glomeruli of the NOD mice. Glomerulosclerosis was caused by excessive fibronectin accumulation. The NOD mice exhibited impairments in renal function caused by elevated serum creatinine and proteinuria levels. These manifestations in the kidneys of the NOD mice were similar to those in the kidneys of the mice with streptozotocin-induced diabetes, which involved chemically induced pancreatic insulitis [38].

To determine whether the cMDSCs developed from the GM-CSF, IL-6, and IL-1β cytokines regulating T-cell activity influence the pathogenesis of T1D in mice, T cells were harvested from the spleens of the NOD mice and administered by themselves or cotransplanted with cMDSCs into NOD–SCID mice. This study indicates that cMDSC administration improved the hyperglycemic state of the NOD–SCID mice induced by the reactive splenic T cells harvested from the NOD mice and prolonged their diabetes-free survival. The lethal complications of diabetes include kidney failure, heart disease, and brain damage. The limitation of this study is that it only focused on investigating the effects of cMDSC application in the pancreas and kidney. Future studies need to be conducted with experiments on cMDSC administration in the heart and brain to elucidate the comprehensive cause of diabetic mortality. Cytokine-induced MDSCs exhibited the immunoregulatory effect of inhibiting effector T-cell function, improved pancreatic insulitis, and increased insulin production. In addition, the application of cMDSCs reduced fibronectin production in the renal glomeruli and improved renal function in the NOD–SCID mice. Our previous study also reported that the anti-inflammatory properties of cMDSCs are crucial for improving renal glomerulosclerosis in mice with chemically induced diabetes [38].

## 5. Conclusions

The administration of cMDSCs propagated from GM-CSF, IL-6, and IL-1β cytokines prolonged the diabetes-free survival of diabetic mice by way of inhibition of activated T cells. In addition, cMDSC use reduced fibronectin production in the renal glomeruli and improved pancreatic insulitis in the NOD-SCID mice. The present study provides an alternative immunotherapy protocol for the treatment of diabetic pancreatic insulitis and renal nephropathy.

## Figures and Tables

**Figure 1 cells-12-01507-f001:**
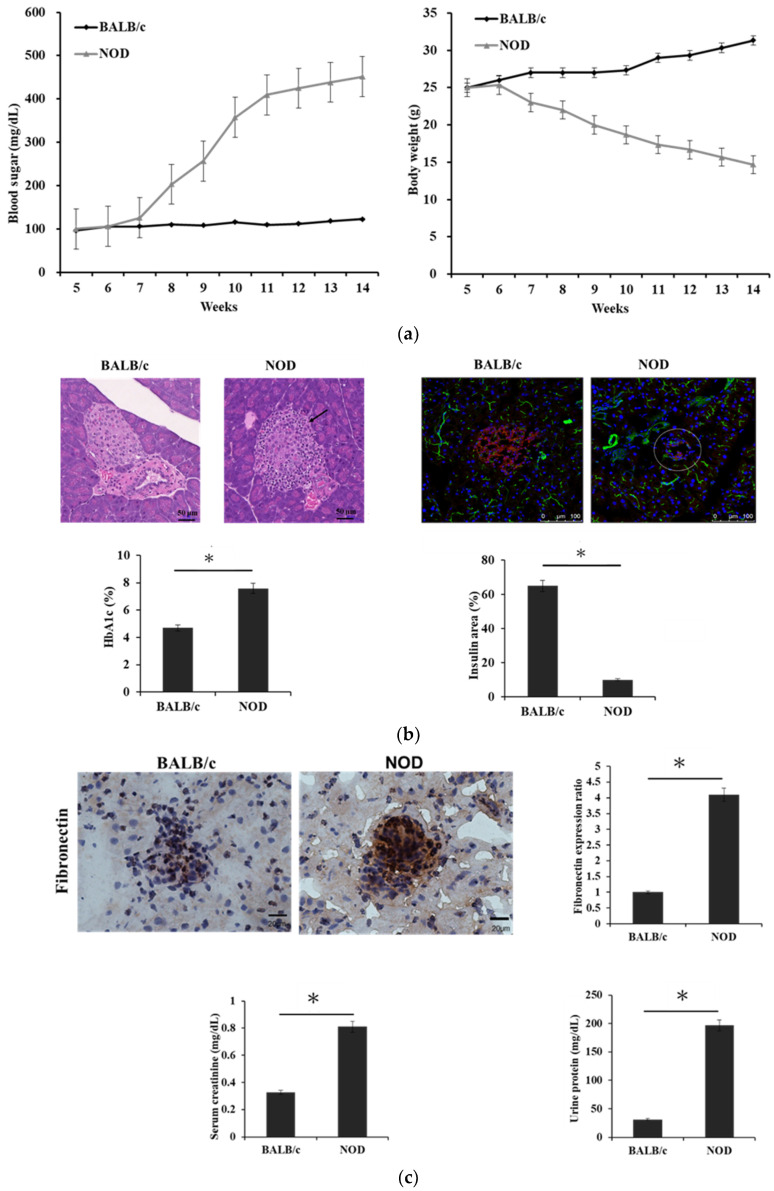
Expression of pancreatic and renal functions in nonobese diabetic (NOD) mice. (**a**) Blood sugar levels (left panel) and body weight (right panel) were measured once per week for the 5-week-old subgroups of BALB/c and NOD mice. Mice in each group were sacrificed on week 14. Diabetes was diagnosed as a sustained blood glucose level >350 mg/dL. (**b**) Pancreas histology for 14-week-old BALB/c and NOD mice stained with hematoxylin and eosin. BALB/c mice exhibited normal pancreatic islets surrounded by an exocrine portion. NOD mice exhibited islet cell destruction with the presence of leukocytes (arrowhead, 400× magnification; upper-left panel). Cryostat sections of the pancreatic islets of the BALB/c and the NOD mice were stained with anti-insulin monoclonal antibody (mAb, red) and evaluated under a fluorescence microscope (400× magnification; upper-right panel). Bar graphs present the blood glycosylated hemoglobin (HbA1c) levels (%) and the insulin area of the pancreatic islets (%) from the BALB/c and the NOD mice (lower panel, * *p* < 0.05). (**c**) Cryostat sections of kidney histochemically stained with anti-fibronectin mAb (upper-left panel, brown; 400× magnification) and examined using a microscope. The ratio of fibronectin expression within the renal glomeruli of the two groups was quantified (upper-right panel, * *p* < 0.05). Bar graphs display the levels of serum creatinine and urinary protein collected from the two groups at 14 weeks (lower panel, * *p* < 0.05). Data are representative of three separate experiments.

**Figure 2 cells-12-01507-f002:**
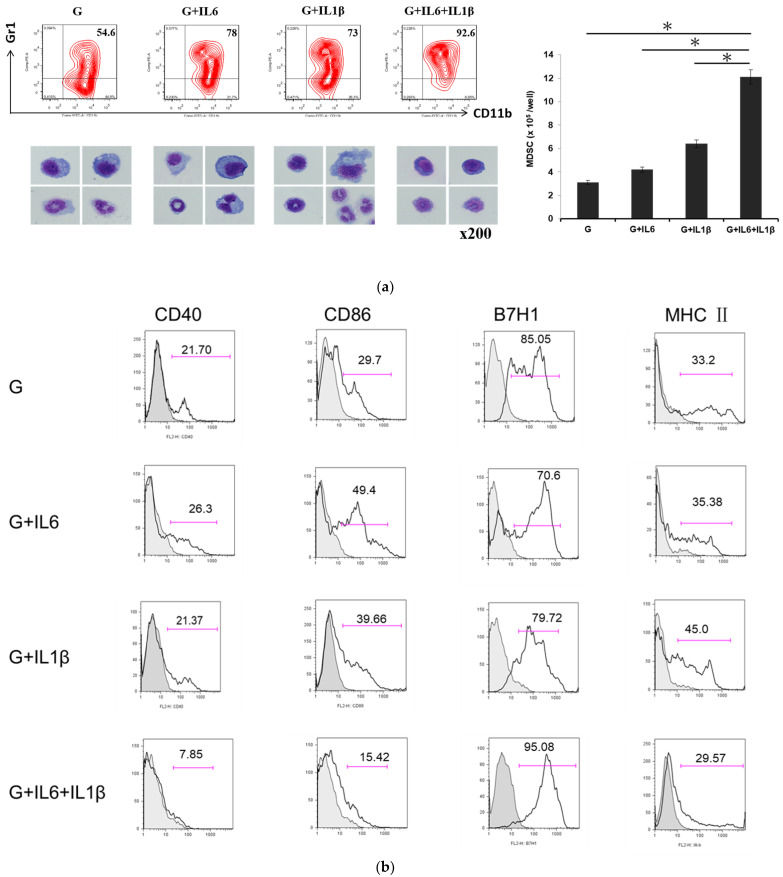
Characterizations of cytokine-induced myeloid-derived suppressive cells (MDSCs). (**a**) MDSCs were propagated from bone marrow (BM) cells (2 × 10^6^ cells/well), harvested from the tibias and femurs of the BALB/C mice, cultured with mouse recombinant granulocyte–macrophage-colony-stimulating factor (GM-CSF; 10 ng/mL), interleukin (IL)-6 (10 ng/mL), and IL-1β (10 ng/mL) for 7 days. MDSCs were diagnosed as double-positive for CD11b and Gr-1. The results are displayed as a histogram in which the numbers correspond to the percentage of positive cells. The MDSCs were stained with Giemsa for morphology examination (200× magnification). The bar graph displays the yield of the MDSCs for each group (* *p*  <  0.05). (**b**) Surface markers of MDSCs stained with specific mAbs against CD40, CD86, B7H1, and MHC class II. Numbers indicate the percentage of positive cells; grey shapes as isotype. (**c**) Expression of the signal transducer and activator of transcription (STAT) 1, STAT3, arginase 1, and inducible nitric oxide synthase (iNOS) mRNA from the MDSCs of each group were determined using a quantitative polymerase chain reaction (* *p*  <  0.05; NS: not significant). (**d**) The antigen-presenting activity of MDSCs. The MDSCs were cultured with B6 spleen T cells labeled with carboxyfluorescein diacetate succinimidyl ester (CFSE) at a ratio of 1:2 or 1:4 for 3 days. Proliferative response was determined through CFSE dilution. (**e**) The levels of IFNγ from T cells were analyzed from the supernatant of MDSC-stimulated T cells through ELISA (* *p*  <  0.05). Data are representative of three separate experiments.

**Figure 3 cells-12-01507-f003:**
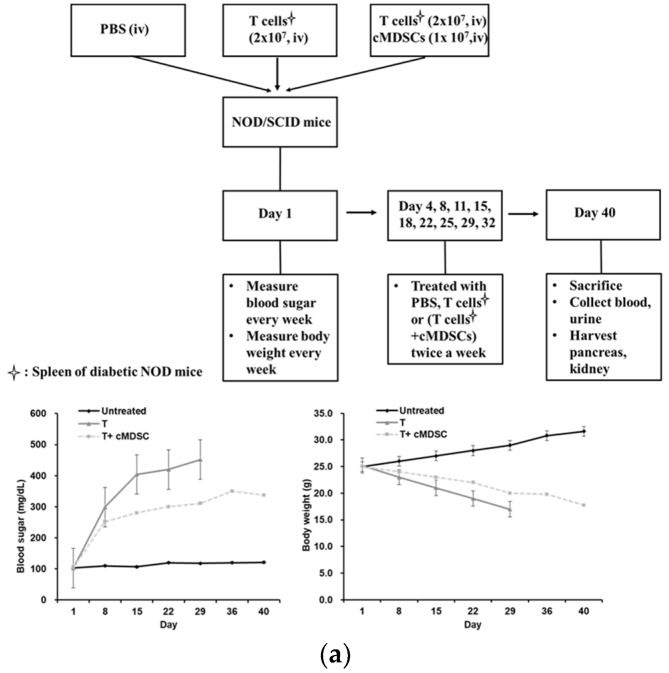
The ability of cMDSCs to prolong the diabetes-free survival of NOD mice with severe combined immune deficiency (SCID) in vivo. (**a**) T cells (CD3^+^, 2 × 10^7^ cells) harvested from the spleens of NOD mice were injected intravenously into NOD–SCID mice with or without cMDSCs (1 × 10^7^ cells) twice a week for 5 consecutive weeks. The control group was treated with phosphate-buffered saline (PBS). Blood sugar levels and body weight (lower panel) were measured every week. The mice were sacrificed on day 40, and blood, urine, spleens, and pancreases were harvested for examination. (**b**) Diabetes-free survival of NOD–SCID mice after treatment with T cells alone (*n* = 5), T cells cotransplanted with cMDSCs (*n* = 5), or PBS (*n* = 5). Diabetes was diagnosed as a sustained blood sugar level > 350 mg/dL (* *p*  <  0.05). (**c**) MDSC ratios in blood, spleen, kidney, and pancreas of three groups compared. Isolated cells were stained with two colors with specific mAbs against CD11b and Gr-1 for flow analyses (* *p*  <  0.05). (**d**) Kidney cryostat sections histochemically stained with antifibronectin (brown, 400× magnification) were examined under a microscope. The area of fibronectin expression within the glomeruli of the three groups was quantified (* *p* < 0.05). The three groups’ serum creatinine and urinary protein levels were examined and compared (* *p* < 0.05). (**e**) Cryostat sections of the pancreas were histochemically stained with anti-insulin antibody (brown, 100× magnification) and examined under a microscope. The area of insulin secretion within the pancreatic islets of the three groups was quantified. HbA1c levels (%) in the blood were measured and compared (* *p* < 0.05). (**f**) The expression of T cells and MDSCs in the pancreatic islets was determined through histochemical staining with anti-CD3 or anti-Gr-1 mAb (brown, 400× magnification) and examination under a microscope (left panel). Bar graphs illustrate positive T cells and MDSC counts in pancreatic tissue as determined using flow analyses (* *p* < 0.05). (**g**) The levels of IFNγ from T cells harvested from pancreatic tissue were analyzed from the supernatant of T cells stimulated with anti-CD3 (1 µg/mL) and anti-CD28 (1 µg/mL) antibodies through ELISA (* *p*  <  0.05). Data are representative of three separate experiments. (**h**) The frameworks of the current studies for the effects of cMDSCs regulated renal and pancreatic inflammations to prolong diabetic mice survival; BMC: bone marrow cell.

## Data Availability

Not applicable.

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
