# Peer review of "Cytokine-Induced Myeloid-Derived Suppressor Cells Demonstrate Their Immunoregulatory Functions to Prolong the Survival of Diabetic Mice"

_cells, 2023, doi:10.3390/cells12111507_

Round 1

Reviewer 1 Report

The main question I have is in Fig.3.  Fig 3ashows even after treatment, blood sugar in treated mice is very high and body weight is still significantly decreasing compared to normal mice. All other parameters in 3-d,e and f indicate the mice are still very sick.  despite this, mortality is dramatically decreased.  Explanation for this observation in the manuscript is not clear.  may be mortality is threshold-dependent for all of these factors. 

In the abstract "..state of hyperglycemic in and prolonged..." the word "in' has to be deleted.

Reviewer 2 Report

The manuscript titled “cytokine induced myeloid derived suppressor cells demonstrate their immunoregulatory functions to prolong the survival of diabetic mice” by Li et al., is an excellent study with novel findings. In this study, the authors use MDSC cells differentiated from mouse BM to evaluate its immunoregulatory potential in the autoimmune NOD mice model and evaluate also the recovery from kidney dysfunction. This study has been performed rigorously and it is a well written manuscript. As I was reviewing this manuscript, I came across some necessary queries, that may require further clarification by the authors.

1)      What was the age of the NOD mice when renal dysfunction or fibronectin accumulation was observed in your experiments.

2)      Why did the investigators choose IL-6 and IL-1β along with GM-CSF, when IL-4 is a more standard procedure for MDSC cell differentiation?

3)      Did the authors investigate the T cell differentiation profile after treatment with MDSCs to see if they were expressing cytokines or markers for Th1, Th2, Th17, or Treg cells?

4)      In your experiments, you have shown that co-transplantation of cMDSCs with T cells has some benefit in reducing renal dysfunction, but in clinical cases, renal dysfunction is diagnosed late, in such cases, it would be nice if there was a model or reversal by cMDSCs to NOD mice that already have increase fibronectin expression. Is this a reasonable experimental group, if yes, why was this not included as a group and what are the limitations of such experimental models.

These were some questions and concerns that if addressed would improve the strength of your manuscript.

The manuscript was well-written with very few grammatical and typographical errors. 
